# Nanoscale Vacuum Diode Based on Thermionic Emission for High Temperature Operation

**DOI:** 10.3390/mi12070729

**Published:** 2021-06-22

**Authors:** Zhihua Shen, Qiaoning Li, Xiao Wang, Jinshou Tian, Shengli Wu

**Affiliations:** 1School of Electronics and Information Engineering, Nantong Vocational University, Nantong 226007, China; 9000082@mail.ntvu.edu.cn; 2School of Electronic Information and Artificial Intelligence, Shaanxi University of Science and Technology, Xi’an 710049, China; 3State Key Laboratory of Transient Optics and Photonics, Xi’an Institute of Optics and Precision Mechanics of CAS, Xi’an 710119, China; tianjs@opt.ac.cn; 4Key Laboratory for Physical Electronics and Devices of the Ministry of Education, Xi’an Jiaotong University, Xi’an 710049, China; slwu@mail.xjtu.edu.cn

**Keywords:** finite integration technique (FIT), thermionic emission, nanoscale vacuum diode, space-charge limited (SCL) current

## Abstract

Vacuum diodes, based on field emission mechanisms, demonstrate a superior performance in high-temperature operations compared to solid-state devices. However, when considering low operating voltage and continuous miniaturization, the cathode is usually made into a tip structure and the gap between cathode and anode is reduced to a nanoscale. This greatly increases the difficulty of preparation and makes it difficult to ensure fabrication consistency. Here, a metal-insulator-semiconductor (MIS) structural nanoscale vacuum diode, based on thermionic emission, was numerically studied. The results indicate that this device can operate at a stable level in a wide range of temperatures, at around 600 degrees Kelvin above 260 K at 0.2 V voltage bias. Moreover, unlike the conventional vacuum diodes working in field emission regime where the emission current is extremely sensitive to the gap-width between the cathode and the anode, the emission current of the proposed diode shows a weak correlation to the gap-width. These features make this diode a promising alternative to vacuum electronics for large-scale production and harsh environmental applications.

## 1. Introduction

As a key component of electronic devices, vacuum tubes were first used in extensive applications such as electron emitters, rectifiers, switches and detectors. However, with the continuous demands of low power consumption and miniaturization, vacuum tubes have been replaced by solid-state electronic devices in most of their applications [1]. The solid-state devices can be easily scaled down by photolithography technique and are compatible with a modern integrated circuit (IC) in a standard semiconductor process. In contrast, vacuum electronics possess intrinsic advantages, as electrons transport ballistically in the vacuum channel while suffering from scattering and collision in solid-state devices. Thus, vacuum electronics may output with high frequency [2,3], an on/off ratio [4] and a fast temporal response [5]. Furthermore, vacuum electronic devices are recognized to be more robust in hostile environments, such as those with a high temperature and ionizing radiation [6].

Efforts have been made to miniaturize vacuum electronic devices to nanoscale dimension, to make them compatible with integrate circuits, and to reduce their power consumption to an acceptable level for modern microcircuits. In order to achieve ballistic transport under low voltage bias, devices with a nanoscale gap were proposed in a number of studies whereby unique fabrication techniques were adopted. These included mechanically break junctions [7], gold plating [8], transmission electron microscope (TEM) [9], focused ion beam (FIB) [10,11,12] and electron beam lithography (EBL) [13,14,15]. In practical applications, sophisticated treatments are needed to achieve a homogeneous gap-width of these devices in order to ensure consistent current density, as the emission current is extremely sensitive to the gap-width according to the tunneling effect [12]. The vertical aligned structure is an alternative method to fabricate the nanoscale gap whereby gap-width can be precisely controlled by the thickness of dielectric layer [16,17,18,19]. In contrast, the vertical structure may be superior in device manufacturing and characteristic consistency to the lateral structure. However, both devices are normally subjected to the tunneling law, where the emission current shows an exponential dependency on gap-width or microscopic morphology of the emitter (field enhancement factor) [20], which poses a serious challenge to fabricating consistent devices during large-scale production.

In [16] and [18], thermionic emission was observed under the low voltage bias of the vertical structural diode. Nevertheless, no further research has been carried out on the temperature-dependent behavior of this kind of device. Here, we report a metal-insulator-semiconductor (MIS) nanoscale vacuum diode based on thermionic emission. The finite integration technique was adopted to numerically study the performance of the diode. The results indicated that the emission current exhibits a 3/2 power dependency of voltage under low bias, similar to the Child–Langmuir law. However, unlike the Child–Langmuir diode, in which two parallel electrodes where emission current has a linear dependency with 1/d^2^ (d is the distance between electrodes) [21], the emission current of the proposed diode shows a weak correlation with the distance. This will benefit the consistency of device performance during large-scale fabrication. Furthermore, as long as the temperature is beyond a certain level, where space charge effect dominates, the emission current remains constant when temperature rises. Thus, the thermionic emission diode is suitable for high-temperature operations. These conclusions may be helpful in accelerating the application of such devices.

## 2. Device Structure and Working Principle

Figure 1a shows the schematic of the diode we proposed, which was a typical metal-insulator-semiconductor (MIS) structure with a nano void channel in the center. The radius of the void channel was 30 μm and the length was set to be less than 100 nm, in order to support ballistic transport. Under the external voltage bias, as shown in Figure 1b, electrons would accumulate in the potential well at the metal–oxide interface on the metal side and confined in a flat space (<1 nm), forming a quasi-two-dimensional electron system (2DES) [16]. In the 2DES, electrons can be considered as free to move in the lateral dimension but forbidden to move in the vertical dimension. Considering the 2DES with a finite lateral extent, electrons near the cleaved edge suffer strong Coulombic repulsion from the internal electrons and are readily injected into vacuum. The potential energy of the edge electrons is higher than that of the internal electrons, or even higher than the surface vacuum barrier, as shown in Figure 1c. These electrons can overcome the surface vacuum barrier and be emitted into the vacuum, and subjected to the thermionic emission law under relatively low voltage bias.

## 3. Methods

An electromagnetic field and particle tracking simulation software, based on a finite integration technique named CST PARTICLE STUDIO^TM^ (a part of CST STUDIO SUITE^TM^ 2012, CST—Computer Simulation Technology, Darmstadt, Germany), were used to numerically explore the performance of nanoscale vacuum diode. The electron emission area, which is related to the thickness of 2DES, was set as 1 nm. The minimum mesh step was set as 0.2 nm. Richardson–Dushman equation, j = AT^2^exp(−Φ/kT), was adopted to calculate the thermionic emission current. In the equation, emission constant A was set as the default value (1.2 × 10^6^A·m^−2^·K^−2^). Φ is the equivalent work function of the cathode, which is much lower than normal values (4.28 eV) due to the Coulombic repulsion in 2DES and needs to be determined in advance, using experimental data. A nanoscale vacuum diode with the exact same geometrical dimension and cathode material as the simulation model was fabricated and tested by Keithley 4200 under room temperature in our former work [17]. The fabrication details and more measured results can be seen in the appended Appendix A.

## 4. Results and Discussion

The measured I–V curve showed two different slopes in a log–log plot, as shown in the inset of Figure 2a, indicating two different electron emission mechanisms. Under a low voltage bias, the emission current exhibited a 3/2 power dependency of voltage, which was essentially governed by the space-charge or the so-called virtual cathode. The potential value and position of the virtual cathode was determined by the anode voltage. A portion of the hot electrons which were emitted would bounce back to the cathode by the virtual cathode. As the voltage increased, the virtual cathode came closer to the cathode and would eventually land on it. As the voltage increased, the emission current then reached a saturation state, since all the hot electrons emitted could be transported in the vacuum channel and eventually be captured by the anode. Actually, the emission current would keep increasing due to the lower barrier height by external electric field on the emission surface (The Schottky Effect). This explained the second part of the curve. The location of the intersectional point of the two different mechanisms was controlled by thermionic emission current density, which was related to the equivalent work function and temperature, as shown in Figure 2a,b. As the I–V curve was measured at room temperature, the equivalent work function could be identified by varying the Φ in simulation to match the inflection point of measured curve, as shown in Figure 2a. Herein, the calculated equivalent work function of the cathode was 0.29 eV. It should be noted that, for simplicity, the surface vacuum barrier lower effect was intentionally ignored in the simulation at a high voltage bias.

Unlike most solid-state diodes, which are sensitive to temperature, the current of the proposed vacuum diode remains constant under different working temperatures with a certain voltage bias in space-charge-limited (SCL) regimes, as shown in Figure 3. When the temperature was relatively low, a small number of electrons were emitted into the vacuum, according to the thermionic emission mechanism. The electric field was able to attract the electrons to the anode in time to prevent them from clogging up in the vacuum channel and forming a virtual cathode. In this case, the virtual cathode could be considered as locating on the cathode. Thus, the emitted electrons were always in an acceleration field once ejected from the cathode. Current is a monotonic function of temperature in accord with the Richardson–Dushman equation. As the temperature increased, the population of electrons finally became too large to be captured by the anode in time, and the virtual cathode thus formed in the vacuum channel. Electron emission was compliant with the SCL mechanism and saturated with temperature. The low temperature limit of saturation range varied from 260 K to 310 K, in accord with the anode voltage implemented from 0.2 V to 1 V, as depicted in Figure 3. In theory, as long as the physical properties of the layers remained stable, the high temperature limit of saturation range could be very high (e.g., the melting temperature of aluminum is 933 K and could be even higher for other materials).

As mentioned above, the electron emission principle of the conventional nanoscale vacuum diode conforms to the field emission mechanism, and the current has an exponential relationship with the electric field intensity. Thus, the gap-width and microscopic morphology of the cathode significantly affects the emission current. Then, the consistency of device characteristics is difficult to ensure in large-scale production. Devices based on thermionic emission mechanism can effectively address this dilemma. According to Child–Langmuir, the space-charge-limited current derived from Poisson equation, the thermionic emission diode based on a parallel-electrode vacuum capacitance shows a 3/2 power dependency of voltage and linear dependency with 1/d^2^. However, the direct mathematical derivation of the emission current from Poisson equation of the MIS structural nanoscale vacuum diode that we proposed here is very complex due to the non-uniform distribution of electric fields. According to the simulation results, based on finite integration technique, the emission current appears to be almost irrelevant to the gap-width when operating in the SCL regime. The emission current of diodes with different dielectric thickness (40 nm, 50 nm, 60 nm, 70 nm, 80 nm) were calculated as depicted in Figure 4a,b. Additionally, it decreased by 1% as the gap-width (dielectric thickness) increased 10 nm, as shown in Figure 4b, which means that the emission current of the diode based on thermionic emission proposed here shows a much weaker gap-width dependency than the tunneling diode or Child–Langmuir diode. This characteristic will definitely benefit the consistency of device performance during large-scale fabrication.

## 5. Conclusions

A MIS structural nanoscale vacuum diode, based on thermionic emission, was numerically studied in this work. We found that this kind of device is suitable for high-temperature operations with a wide temperature range of around 600 K above 260 K, as long as the physical property of layers remains stable. Thus, as a vacuum electronic device, it is superior to solid-state electronic devices in high-temperature performance. Furthermore, conventional vacuum diodes, commonly based on field emission, need a delicate fabrication process to form a tip cathode or precise control of the cathode-anode gap-width to ensure a consistent field emission of current. However, the emission current of the nanoscale vacuum diode, based on thermionic emission, used here is basically independent of the gap-width. This benefits the easy fabrication of consistent devices for large-scale production. Due to these two valuable traits, this may be identified as a unique and promising device for vacuum electronic applications. Considering that the conclusions are based on pure simulation, experimental verification will be carried out in the subsequent work.

## Figures and Tables

**Figure 1 micromachines-12-00729-f001:**
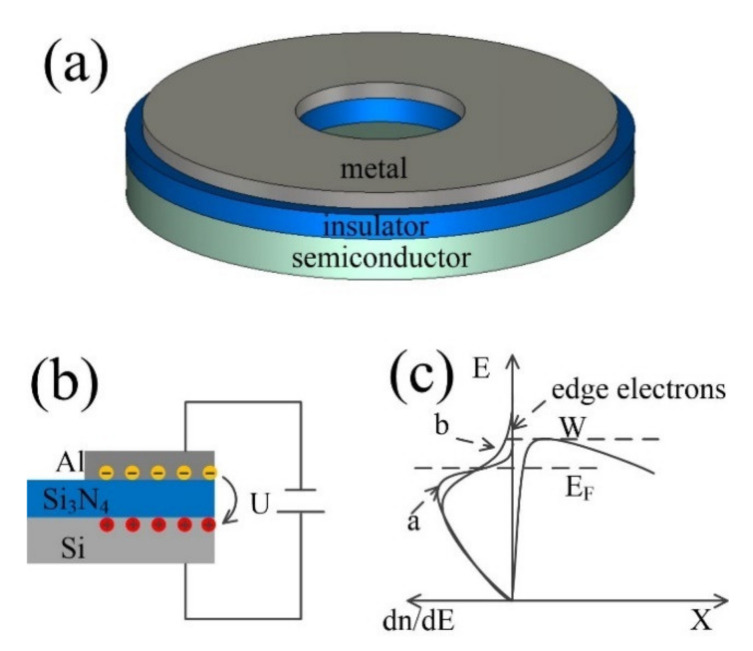
Thermionic emission diode. (**a**) Schematic of the proposed diode; (**b**) Schematic illustration of electrons emission and transport in the vacuum channel. Al was under negative bias. Electrons were ejected from the edge of 2DES travelling from metal to silicon via vacuum channel; (**c**) Diagram of electron distribution by energy in cathode and surface vacuum barrier. W was the height of surface vacuum barrier under certain bias. EF was the Fermi level of cathode. Line a was normal electron distribution in metal, which we deliberately exhibited here as a reference. Line b was electron distribution in metal where 2DES developed under certain bias.

**Figure 2 micromachines-12-00729-f002:**
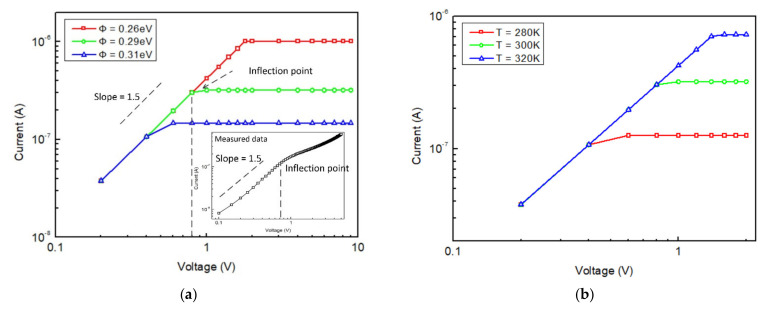
(**a**) Measured I-V curve and simulated I-V curve with different equivalent work function; (**b**) Simulated I–V curve under different temperature where the equivalent work function of cathode was set as 0.29 eV.

**Figure 3 micromachines-12-00729-f003:**
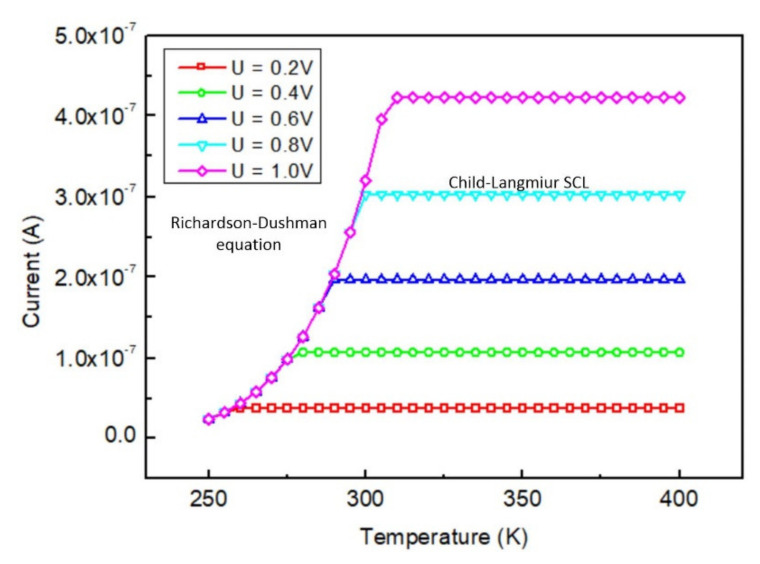
Simulated current versus temperature under different voltage bias.

**Figure 4 micromachines-12-00729-f004:**
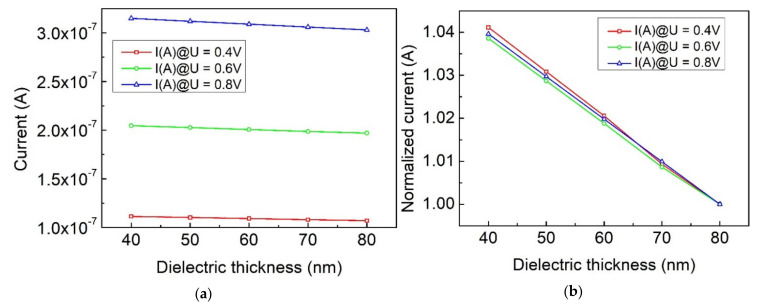
(**a**) Simulated emission current versus gap-width (dielectric thickness) under different voltage bias. (**b**) Normalized emission current versus gap-width (dielectric thickness) under different voltage bias.

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
