# Peer review of "Nanoscale Vacuum Diode Based on Thermionic Emission for High Temperature Operation"

_micromachines, 2021, doi:10.3390/mi12070729_

Round 1

Reviewer 1 Report

No significant improvement or new science on this manuscript compared to the previous reports. Just by simulating the device with several different temperature without significant development is not enough for the publications.  

Reviewer 2 Report

Title: Nanoscale Vacuum Diode Based On Thermionic Emission For High Temperature Operation And Large-scale Production
Authors: Zhihua Shen, et al.
Micromachines 2021 Manuscript No. 1235268

A metal-insulator-semiconductor (MIS) nanoscale vacuum diode based on thermionic emission was designed to achieve ballistic transport under low voltage bias. This device was simulated under a wide range of temperatures revealing potentially valuable characteristics. 

This manuscript consists of a non-structured abstract with keywords, 5 sections (introduction, device structure and working principle, results and discussion, and conclusions) on 6 pages of single-spaced text with embedded figures. There are 21 references, 4 figures, and no tables. No appendices, supplements, or URL are provided.  

A previously described metal-insulator-semiconductor (MIS) vacuum diode based on thermionic emission was evaluated for high temperature performance using numerical simulation. It is not clear that this device has been fabricated and tested. This MIS vacuum diode has significant potential for improved high temperature operation and simplified fabrication.    

MIS is not defined in the abstract. (line 18) 

This paper has no "Methods" section. The results and discussion sections are combined. It would be valuable to separate these sections and include distinct "Methods", "Results", and "Discussion" sections. 

Although this paper is focused on numerical simulation, Figure 1a shows a "Measured I-V curve". The apparatus and setup used to make these measurements are not described in detail. How were the measurements obtained?

Although there is no "Methods" section, the 1st paragraph of the "Results and discussion" section describes the use of CST particle studio software. No specific reference for this software is cited. I presume that the authors refer to CST PARTICLE STUDIO® (CST PS), a specialist tool for the fast and accurate analysis of charged particle dynamics in 3D electromagnetic fields. Please clarify.

The platform used for numerical simulation using CST particle studio is not mentioned. Which version of this software was used? A detailed list of parameters used in the simulation is not available. The authors mention that "Fabrication details were illustrated in the Supplementary Information of our former work [17]." This reference, #17, and the accompanying supplement are not available without a subscription to "Vacuum". It would be preferable to update this supplement and attach it to the current manuscript. 

"Thermionic Emission in a Planar Diode" is available in COMSOL Multiphysics 5.6. Why was CST particle studio chosen? Was COMSOL considered as an alternative?

As a result of numerical simulation, two important characteristics of the device under test were found: (1) reduced need for high precision fabrication, and (2) independence of emission current from gap width. The most important requirement for future work is experimental verification of these characteristics. No statement regarding plans for future work is included. Why not?

Overall, this report describes numerical simulation of an innovative MIS vacuum diode that employs thermionic emission revealing potentially valuable properties in high temperature operation and device fabrication. The manuscript is incomplete without an accompanying supplement. Separation of "methods", "results", and "discussion" sections is recommended. Since the report is devoted to simulation, complete details on how this was implemented are required. It would not be possible to reproduce this work given the brevity and omission in the current report. Recommendations for future work, especially to perform experimental verification of the findings, is an important addition to the current report. 

Reviewer 3 Report

This work reports a numerical study of the thermionic emission under low voltage bias of a vacuum diode based on vertical aligned structure. The emission current dependency on voltage, on the working temperature and on the gap width between cathode and anode are characterized. The results are interesting, and probably the paper is worth publishing, though the writing would benefit from improvement and revision. There are sentences which have grammar issues and/or are an unclear.

specific comments:

1) The words “large-scale production” should be removed from the title since they are improper, while the concept can be left in the manuscript.

2) The stability of the device up to 933K has not been demonstrated numerically in the manuscript, it is only an assumption. The related sentences in the abstract and in the main text should be relaxed accordingly.

3) The explanation of the reason why the emission current of your vacuum diode device is almost irrelevant to the gap width should be suggested and further discussed in the manuscript.

4) Lines 18-19 have to be rewritten in a clearer form. The acronym of MIS has to be inserted.

5) Lines 36-37 contain two while and have to be rewritten.

6) Lines 71-72: last sentence is written in a too personal mode. Rewrite it into a professional form.

7) Line 138. A dot has been forgotten

Round 2

Reviewer 1 Report

They added little more explanation without improving/adding any work. As I mentioned in my previous comments, this manuscript does not provide any improvement of last development in this field. This study could be interesting with experimental work but authors decided to do the experiment later. In my opinion, manuscript as it is does not entertain the reader compared to the previous studies. 

Reviewer 2 Report

Title: Nanoscale Vacuum Diode Based On Thermionic Emission For High Temperature Operation
Authors: Zhihua Shen, et al.
Manuscript No. micromachines-1235268 

A revised manuscript accompanied by a point-by-point response to the initial review (#2) contains significant changes. The revised version clarifies and expands on the methods used in this simulation study. The addition of a supplement with 3 paragraphs of text and 2 additional figures is a major improvement.

Reorganization of this manuscript with separation of the "Methods" greatly improves the readability. 

This revised manuscript, augmented with "Supplementary Materials", does address all of the questions and concerns raised in the initial critique. The conclusions are supported by the data presented in this report. The authors state that they plan to pursue fabrication and experimental testing of a nanoscale vacuum diode to be reported later. 

This work builds upon the authors' prior work (ref. 17) from 2017 with significant improvement in high temperature operation demonstrated by simulation using CSI Particle Studio. The report is well organized, illustrated, and complete. 
